# Diversity of Mitochondrial DNA Haplogroups and Their Association with Bovine Antral Follicle Count

**DOI:** 10.3390/ani12182350

**Published:** 2022-09-08

**Authors:** Hongfei Liu, Junjun Zhai, Hui Wu, Jingyi Wang, Shaowei Zhang, Jie Li, Zhihan Niu, Chenglong Shen, Kaijuan Zhang, Zhengqing Liu, Fugui Jiang, Enliang Song, Xiuzhu Sun, Yongsheng Wang, Xianyong Lan

**Affiliations:** 1College of Animal Science and Technology, Northwest A&F University, Yangling 712100, China; 2College of Life Science, Yulin University, Yulin 719000, China; 3College of Veterinary Medicine, Northwest A&F University, Yangling 712100, China; 4Shandong Key Laboratory of Animal Disease Control and Breeding, Institute of Animal Science and Veterinary Medicine, Shandong Academy of Agricultural Sciences, Jinan 250000, China; 5College of Grassland Agriculture, Northwest A&F University, Yangling 712100, China

**Keywords:** bovine, mitochondrial DNA haplogroup, maternal origin, ovary, antral follicles

## Abstract

**Simple Summary:**

Chinese Holstein cows, crossbred cattle in China, have abundant genetic diversity due to the fact that Holstein cows were crossbred with various local breeds. Therefore, to reveal their genetic diversity, we sequenced the complete mitochondrial DNA (mtDNA) D-loop region of 501 Chinese Holstein cows and found abundant genetic resources (including 219 haplotypes, polymorphisms of 260 SNPs, and 32 indels) and two haplogroups in the study population. Meanwhile, the two haplogroups had different maternal origins and corresponding distinguished mutations. Furthermore, the association between haplogroups and antral follicle count was found, and the mtDNA haplogroup HG1 with Bos indicus maternal origin had more antral follicles (diameter ≥ 8 mm), implying that these individuals of HG1 had higher reproductive potential. Finally, these distinguished mutations between two haplogroups could be considered as important genetic markers for cattle breeding.

**Abstract:**

Maternal origins based on the bovine mitochondrial D-loop region are proven to have two main origins: *Bos taurus* and *Bos indicus*. To examine the association between the maternal origins of bovine and reproductive traits, the complete mitochondrial D-loop region sequences from 501 Chinese Holstein cows and 94 individuals of other breeds were analyzed. Based on the results obtained from the haplotype analysis, 260 SNPs (single nucleotide polymorphism), 32 indels (insertion/deletion), and 219 haplotypes were identified. Moreover, the nucleotide diversity (π) and haplotype diversity (Hd) were 0.024 ± 0.001 and 0.9794 ± 0.003, respectively, indicating the abundance of genetic resources in Chinese Holstein cows. The results of the median-joining network analysis showed two haplogroups (HG, including HG1 and HG2) that diverged in genetic distance. Furthermore, the two haplogroups were significantly (*p* < 0.05) correlated with the antral follicle (diameter ≥ 8 mm) count, and HG1 individuals had more antral follicles than HG2 individuals, suggesting that these different genetic variants between HG1 and HG2 correlate with reproductive traits. The construction of a neighbor-joining phylogenetic tree and principal component analysis also revealed two main clades (HG1 and HG2) with different maternal origins: *Bos indicus* and *Bos taurus*, respectively. Therefore, HG1 originating from the maternal ancestors of *Bos indicus* may have a greater reproductive performance, and potential genetic variants discovered may promote the breeding process in the cattle industry.

## 1. Introduction

The mitochondrial genome (mitogenome) of cattle is a double-stranded DNA sequence of 16,338 bp (reference mitochondrial DNA: NC_006853.1). It shares high homology with the human mitogenome [1], containing 12S and 16S rRNA, 22 tRNAs, some protein-coding genes, and a non-coding region (NCR) [2]. The D-loop region, a triple-stranded region in the NCR, is the most divergent region in mammalian mitogenomes [2]. However, the middle of the D-loop region is more conserved than the extremities. Domestic cattle’s maternal origins and phylogenetic structure based on maternally inherited mitogenomes have been studied extensively worldwide, especially the D-loop region sequence of mitochondrial DNA (mtDNA), which is the main research subject owing to its highly variable pattern. Modern cattle are derived from wild ancestral aurochs (*Bos primigenius*) and have two main maternal origins, *Bos taurus* (T haplogroup) and *Bos indicus* (I haplogroup) [3]. A study of the mtDNA sequences of modern Asian cattle from six Asian countries indicated two major haplogroups (HG): T and I [4]. The T haplogroup could be divided into four main subgroups: T1A, T2, T3 (containing T3A and T3B), and T5. The I haplogroup can be divided into I1 and I2 [5]. Even though the T4 haplogroup is a novel group with a (T > C) mutation [5], it can also be embedded within the T3A haplogroup because of the closer genetic distance. Other rare haplotypes (P, Q, R, C, and E) were also identified in specific breeds of modern cattle or aurochs, but haplotype P is infrequent in Asian cattle [3].

Chinese Holstein cows were first bred by crossing Holstein–Friesian cows with various native cattle breeds in China approximately 150 years ago [6]. Therefore, Chinese Holstein cows have a rich genetic resource that deserves to be discovered and explored. After the introduction of Holstein cows, the divergence of local Chinese Holstein cows gradually decreased because of the exchange of genetic resources between the different districts. However, many Chinese Holstein cows’ reproductive performance varies considerably [7]. Furthermore, almost all cattle breeds in China can be classified into three groups based on geographic location and mtDNA maternal origin: the *Bos taurus* haplogroup in Northern China, the *Bos indicus* haplogroup in Southern China, and the hybrid haplogroup in the central area of the Yellow River [8,9,10]. Moreover, as the geographic distribution indicates, there is a decreasing gradient of zebu introgression from south to north [11,12]. Therefore, Chinese Holstein cows in different districts may also have similar geographical distributions and can be divided into two major haplogroups: *Bos taurus* and *Bos indicus* haplogroups. However, the relationship between haplogroups and reproductive traits remains unclear.

Specific mutations or haplotypes in mtDNA have been widely reported and are proposed to be associated with bovine reproductive traits [13,14]. Some studies have demonstrated that the divergence of mtDNA copy number exists between *Bos taurus* and *Bos indicus* [15] and is related to oocyte quality [15]. In addition, a positive relationship between oocyte quality and antral follicle count (AFC) in Brahman (*Bos taurus indicus*) and Simmental cows (*Bos taurus taurus*) has been reported [16]. Meanwhile, researchers found that AFC was larger in Brahman cows than in Simmental cows and was positively related to the oocyte quality, embryo number, and pregnancy rate [16], thus suggesting the superior reproductive performance of *Bos indicus*. Therefore, it is hypothesized that genetic diversity of mtDNA haplotypes exists in Chinese Holstein cows and that different haplogroups may be associated with reproductive traits. We then focused on ovarian morphological traits and antral follicle count, one of the crucial reproductive performance indicators [17,18,19], to analyze mtDNA D-loop region polymorphisms in the ovaries of 501 Chinese Holstein cows from the Guanzhong Plain.

## 2. Materials and Methods

Animal experiments were performed according to the Regulations for the Administration of Affairs Concerning Experimental Animals (Ministry of Science and Technology, China, 2004) and were supported by the Welfare Committee of Northwest A&F University (IACUCNWAFU; protocol number NWAFAC1008).

### 2.1. Samples and Reproductive Traits Records

In this study, all samples (*n* = 501) of Chinese Holstein cows were randomly collected from different individuals’ left or right normal ovaries at a slaughterhouse in Xi’an in autumn and stored at −80 °C. Considering the random sampling process, the potential effect of family structure and other genes were ignored in this study. Due to low fertility and breeding arrangements, all adult Chinese Holstein cows (5–6 years old) from the Guanzhong Plain were transported to the slaughterhouse for elimination [20,21,22]. Relevant morphological traits were recorded, including ovarian volumes (derived from the length, width, and height) and weight, the antral follicle count (number) and size (diameter ≥ 8 mm), and the corpus luteum (CL) types [20,21,22]. Morphological phenotypic traits were measured by using a Vernier caliper or a double ruler (Appendix A). The ovary volumes were determined as follows:(1)$$V=\frac{4}{3}\pi abc$$
where V is the volume of each ovary; and a, b, and c refer to half of the size parameters, i.e., length, width, and height, of each ovary [23].

### 2.2. Genomic DNA Extraction and Sequencing

Genomic DNA was extracted from ovarian tissue samples by using the phenol–chloroform method [24]. The primer pairs were designed by using PRIMER PREMIER software (version 5.0; PREMIER Biosoft Co Ltd., San Francisco, California, USA) to detect the complete mtDNA D-loop sequence: (5′-CTATTCCCTGAACACTATTA-3′) and (5′-GAGACTCATCTAGGCATTT-3′). The amplification conditions in this PCR amplification system using a total volume of 50 μL were as follows: 4 min denaturation at 95 °C, 40 cycles of 30 s at 94 °C, 1 min at 59 °C, and 90 s at 72 °C, followed by 10 min at 72 °C as the final extension step. The PCR products from all individuals were directly sequenced without purification, using the above primers (TsingKe Biotech, Xi’an, Shaanxi, China).

### 2.3. Clustering and Haplotype Analysis

All mtDNA D-loop sequences were edited by using the DNAstar package (DNASTAR, Madison, WI, USA) according to the reference sequence of the Korean cattle mtDNA D-loop (NCBI Accession ID: NC_006853.1). Multi-sequence alignment (MSA) was performed based on the ClustalW algorithm in MEGA-X (using default parameters; https://www.megasoftware.net, accessed on 16 November 2019) [25]. Nucleotide/haplotype diversity (π/Hd) and haplotype analysis (excluding indels) were performed by using DNASPv5 (http://www.ub.edu/dnasp, accessed on 1 July 2020) [26]. Construction of the median-joining (MJ) network was based on Popart software (http://popart.otago.ac.nz, accessed on 1 July 2020) [27] for clustering analysis to identify haplogroups.

### 2.4. Phylogenetic Tree Construction

A neighbor-joining (NJ) phylogenetic tree was constructed based on the alignment results of MEGA-X. The number of bootstrap replications was set to 1000 to evaluate the confidence level of each clade in the NJ tree. The NJ tree was rooted in *Bos grunniens* (AY684273.2), which was used as the outgroup. The other parameters were set to their default values in MEGA-X. In addition, the maternal origins of all samples from Chinese Holstein cows, 94 Bos taurus (Asian cattle: Qinchuan cattle, Jinnan cattle, Yunnan yellow cattle, Luxi cattle, Yanbian cattle, Mongolian cattle, and Chinese Holstein cows; European cattle: Angus cattle, Charolais cattle, and Simmental cattle) and *Bos indicus* (Indian zebu) mtDNA D-loop sequences (specific description was available in Table 1) were downloaded from the NCBI database for multi-sequence alignment and construction of NJ tree. Figtree v1.4.3 (https://github.com/rambaut/figtree/releases/tag/v1.4.3, accessed on 28 June 2020) was used to edit and annotate the NJ tree.

### 2.5. Statistical Methods

Based on the results of the MJ network, all samples were divided into two haplogroups, HG1 and HG2. To analyze the association between haplogroups and reproductive traits and to avoid the interference of ovarian weight and types of CL (including conical type, I type; crater type, II type; mushroom type, III type; flat sheet type, IV type; and non-surface type, V type) [28] in Chinese Holstein cows, a two-way co-variance model (ANCOVA) was used with SPSS software (version 23.0; IBM Corporation, Armonk, New York, USA). Meanwhile, the research subject was changed to specific individuals with all valid records (*n* = 115) for achieving high confidence results. The general linear model in ANCOVA was applied as follows: *Y_ijk_* = *μ* + *H_i_* + *S_j_* + *e_ijk_*, where *Y_ijk_* is the adjusted phenotypic values of reproductive traits by the factor of ovarian weight as a covariate, *μ* is the overall adjusted mean of all samples, *H_i_* is the fixed effect of the haplogroup, *S_j_* is the fixed effect of the types of CL, and *e_ijk_* is the random error. As the reproductive traits do not conform to the normal distribution (Shapiro–Wilk test) or variance homogeneity (Levene’s test), the Scheirer–Ray–Hare test [29] (non-parametric form of two-way ANOVA) was used after adjusting dependent variables with significant covariate coefficients (Appendix A).

### 2.6. Principal Component Analysis

The haplotype analysis was carried out according to all the mtDNA D-loop sequences from two haplogroups and other breeds (including datasets of Chinese Holstein cows from the NCBI database). Each variant site in the haplotype was considered a feature that impacted the maternal origin. Subsequently, principal component analysis (PCA) was applied to extract the top two principal components, using plink v1.90 beta (http://www.cog-genomics.org/plink/1.9/, accessed on 2 July 2020). A PCA dot plot was constructed based on the eigenvalues and eigenvectors of the top two components and visualized by using R (package: ggplot2 v3.3.1, New York, USA, https://github.com/tidyverse/ggplot2, accessed on 2 July 2020) software environment.

## 3. Results

### 3.1. Abundant Genetic Variations within the mtDNA D-Loop Region in Chinese Holstein Cows

After multi-sequence alignment of the bovine mtDNA D-loop region, all samples were divided into 219 haplotypes. Additionally, 260 single nucleotide polymorphisms (SNPs) and 32 insertion/deletions (indels) were found, indicating high genetic diversity in Chinese Holstein cows (Appendix A). Based on the statistical analysis of nucleotides and haplotypes, nucleotide (π) and haplotype diversity (Hd) were 0.024 ± 0.001 and 0.9794 ± 0.003, respectively. Moreover, nucleotide divergence reached 17.853, and Tajima’s D values of these variants were less than 0, implying that these variants in the haplotype were influenced by artificial selection or migration.

### 3.2. Distinct Mutations between Haplogroups

The median-joining (MJ) network showed that all haplotypes could be classified into two clusters/haplogroups (HG), HG1 (*n* = 127) and HG2 (*n* = 374) (Figure 1A). Each cluster also had sub-clusters resulting from variants with a lower genetic distance. The distribution of haplogroups was consistent with the maternal origins and corresponding distinctive mutations reported in previous studies [4]. HG1 could be divided into two sub-haplogroups: I1 and I2, which had *Bos indicus* maternal origin. Five sub-haplogroups (T1A, T2, T3A, T3B, and T4) originating from *Bos taurus* maternal ancestors were found in another haplogroup (HG2). The 20 main variants which could be categorized into two clusters were located in the mitochondrial region (reference sequence ID: NC_006853.1): 15951, 15953, 15959, 15994, 16049, 16058, 16082, 16102, 16109, 16116, 16119, 16137, 16138, 16229, 16247, 16248, 106, 166, 169, and 206 (Figure 1B and Appendix A). Other variants also influenced the classification of haplogroups or subclusters, as shown in Appendix A. Construction of the NJ consensus tree also indicated two clades (Figure 2).

### 3.3. Association Analysis of Haplogroups with Antral Follicle Count

After the normal distribution test and evaluation of homogeneity of variances and regression slopes for different reproductive traits, the AFC, ovarian weights, ovarian volumes, and number of CL did not pass the Shapiro–Wilk test or Levene’s test (Appendix A) so that Scheirer–Ray–Hare test was used. Meanwhile, these interaction effects between different fixed factors and ovarian weight in two-way ANCOVA demonstrated that it mainly failed to the homogeneity of regression slopes in AFC and mean diameter of AFC (Appendix A). The results of the Scheirer–Ray–Hare test or two-way ANCOVA indicated that most reproductive traits were not significantly associated with haplogroups (Appendix A). However, antral follicle count was significantly associated with haplogroups (*p* < 0.01) (Table 2 and Appendix A). As shown in Appendix A, the effect of CL type was not significant. In addition, individuals of HG1 with *Bos indicus* maternal origin had a higher antral follicle count (up to 1.55) than that of HG2 in Chinese Holstein cows (1.16).

### 3.4. Different Maternal Origins of HG Haplogroups

The sequencing data composed of the mtDNA D-loop sequences from Chinese Holstein cows combined with a dataset of other breeds were used to carry out MSA and construct the NJ tree for further maternal origin analysis. The NJ phylogenetic tree showed two main clades indicated by different colors (Figure 3; red, HG1; green, HG2; black, *Bos grunniens*). Most Northern China cattle and all European cattle breeds had a closer genetic distance with HG2, but Indian zebu and most Southern China cattle were grouped in the same clade together with HG1. Besides, almost all Mongolian and Luxi cattle (Northern China cattle) had the same maternal origin as HG1.

PCA was also performed to support haplogroups with different maternal origins. By extracting the top two principal components, Indian zebu, Luxi cattle, and Mongolian cattle were still grouped in the same cluster together with HG1 (Figure 4). Furthermore, HG2 had the same maternal origin as the other breeds that mainly originated from *Bos taurus*. However, some individuals from HG2 existed in neither cluster, implying that they correlated with a hybrid maternal origin.

## 4. Discussion

Although an early study explored the haplotype diversity of mtDNA D-loops in Chinese Holstein cows, the evidence supporting the existence of different haplogroups was insufficient (sample size: *n* = 2) compared to this study (*n* = 501) [30]. Furthermore, the present study identified relevant distinctive mutations between the two maternal origins, as is consistent with the results of previous studies [31]. Compared to the reported mutations [31], due to the use of different reference mtDNA sequences (ID: NC_006853.1 vs. V00654), the genomic location of these variants in our study had a 1–3 bp shift. HG1 had the same maternal origin (*Bos indicus*) as Indian zebu, Yunnan yellow cattle, Mongolian cattle, and Luxi cattle, as shown in Figure 3 and Figure 4. Although Mongolian and Luxi cattle are main breeds with *Bos taurus* phenotypic features and are widely distributed in northern China, previous studies have reported the presence of *Bos indicus* haplotypes belonging to the I1 or I2 haplogroup [12], and the introgression event occurred during the second–seventh centuries AD in Mongolian cattle [32]. Finally, we found the association of haplogroups with AFC, which might improve the efficiency of in vitro embryo production (IVP) efficiency through acquiring many mature and high-quality oocytes/follicles from HG1 population. Although the family structure was not involved in this statistical model, the identification and effect evaluation of subpopulation (haplogroup) might account for the partial influence resulting from family structure.

Oocyte quality is associated with copy number variation in mtDNA of different maternal origins [33]. Moreover, the infertility of repeat breeder cows is also influenced by the decrease in mtDNA copy number or differential gene expression between summer and winter [33]. Regarding energy metabolism in mitochondria, some studies indicated that protein-coding gene variants and different haplogroups were associated with the quantity of mtDNA or adenosine triphosphate (ATP) content in vitro [34]. Studies on bovine mtDNA have revealed the impact of different maternal origins on copy number variation and its association with oocyte quality [15]. Similarly, in this study, the relationship between the number of AFCs and haplogroups with different maternal origins was also confirmed. Srirattana et al. [15] reported that oocytes originating from *Bos indicus* are usually of lower quality due to the low mtDNA copy number. However, this study associated the maternal origin of *Bos indicus* with more antral follicles. Moreover, in other relevant studies, it has been reported that the antral follicle count is positively correlated with oocyte quality, embryo number, and pregnancy rate in both Brahman and Simmental cows, and Brahman cows (*Bos taurus indicus*) have a larger AFC and a higher number of viable oocytes than Simmental cows (*Bos taurus taurus*) [16]. The results of another study also reached similar conclusions to this study [35]. As for the ovaries with higher AFC, 17β-estradiol and progesterone ratios, oocyte volumes, and nuclear maturation rates were greater than those in the lower AFC group [35]. Additionally, oocytes in the high AFC group had higher blastocyst development rates (9.1%) [35]. Collectively, the individuals of HG1 with higher AFC may have potential reproductive performance, but it needs to be further verified.

The polymorphic loci of mtDNA have been identified and are associated with reproductive traits in livestock [36]. For example, a significant association was found between the mean live litter size per year and the mtDNA D-loop region in two cattle breeds [36]. This study verified the significant association of AFC with haplogroups maternally originating from *Bos indicus*. We also revealed potential distinctive mutations in the two haplogroups, but the potential molecular mechanism was not further explored, especially for not proving the association of these mutations with mtDNA copy number. However, some evidence on mtDNA copy number and mutations in mtDNA may indicate their associations and possible mechanism. The higher mtDNA copy number possibly caused by these mutations affects energy metabolism and nuclear–mitochondrial interactions in cattle somatic cell nuclear transfer (SCNT) embryos [37], influencing the development of bovine embryos during the blastocyst stage [36]. Of course, considering that these mutations in the mtDNA D-loop region are only partial distinctive markers between two haplogroups, the effect of these mutations may be relatively weaker. Therefore, potential mutations between the two haplogroups affecting the reproductive traits of cows are worthy of further research in the cattle industry.

Based on the results of a genome-wide association study (GWAS), many selective signatures were found in Chinese Holstein cows, which would help understand the mechanism underlying the regulatory roles of genetic variants in cattle breeding [38,39,40,41]. These studies also indicated that genetic resources were abundant and needed to be further examined in Chinese Holstein cows. Furthermore, more extensive research on polymorphisms within specific genes revealed that they correlated with milk production traits and reproduction traits in Chinese Holstein cows [42,43,44,45]. In particular, in studies on mutations within mitochondrial genes [36,46,47,48,49], the association of mtDNA genetic diversity with reproductive traits or environmental adaptation to high altitudes in cattle or yaks was also revealed. Additionally, it has been reported that the intensity of energy metabolism involved in the tricarboxylic acid cycle (TCA) or oxidative phosphorylation (OXPHOS) might be regulated by the copy number of mtDNA [34]. Meanwhile, a difference in the copy number of mtDNA in different haplotypes/haplogroups has also been found [13,15]. This implies that distinctive mutations between different haplogroups with different maternal origins may affect the maturation of follicles by regulating the copy number of energy-metabolism-related genes in mtDNA.

## 5. Conclusions

Chinese Holstein cows had abundant genetic resources result from having two haplogroups with different maternal origins. The mtDNA haplogroup (HG1) with the *Bos indicus* maternal origin exhibited an improved potential reproductive performance due to its having more antral follicles than the mtDNA haplogroup (HG2) with the *Bos taurus* maternal origin.

## Figures and Tables

**Figure 1 animals-12-02350-f001:**
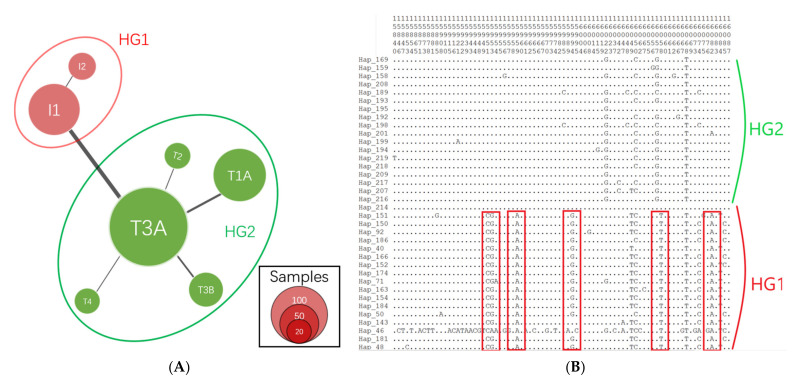
Haplogroups and distinctive mutations in Chinese Holstein cows. (**A**) Median-joining (MJ) network of haplotypes in Chinese Holstein cows. The size of circle represents the number of individuals in this haplotype. Black lines within the branches represent different genetic distance between haplotypes. (**B**) The distinctive polymorphism sites (a portion of all sites) between two haplogroups. Polymorphism sits encircled by red boxes are crucial variants distinguishing two haplogroups (red, HG1; green, HG2). The numbers in each site (top) represent the genomic coordinates in mitogenome.

**Figure 2 animals-12-02350-f002:**
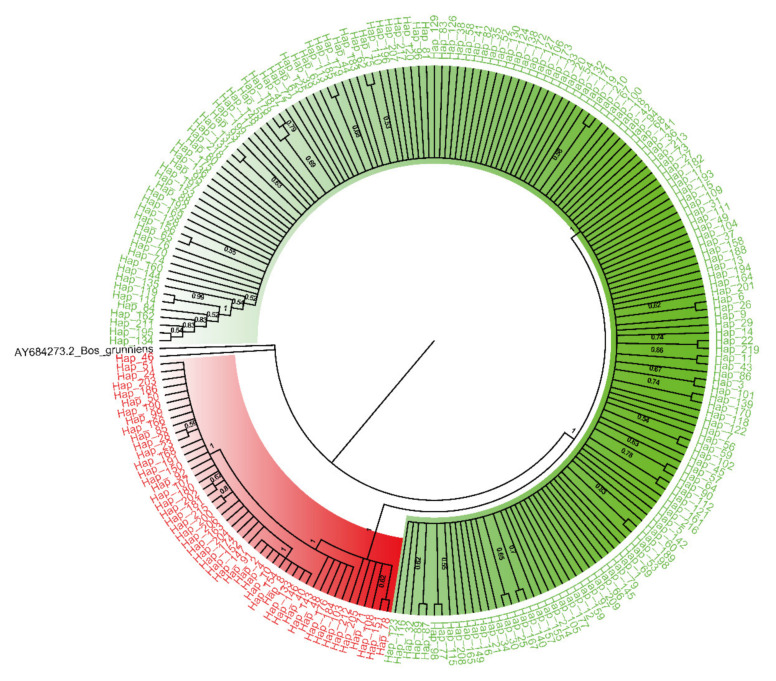
NJ phylogenetic tree of haplotypes of all ovary samples (red, HG1; green, HG2; black, *Bos grunniens*). The values on the branches represent the confidence level (frequencies of occurrence in 1000 replicates) supported by bootstrap method.

**Figure 3 animals-12-02350-f003:**
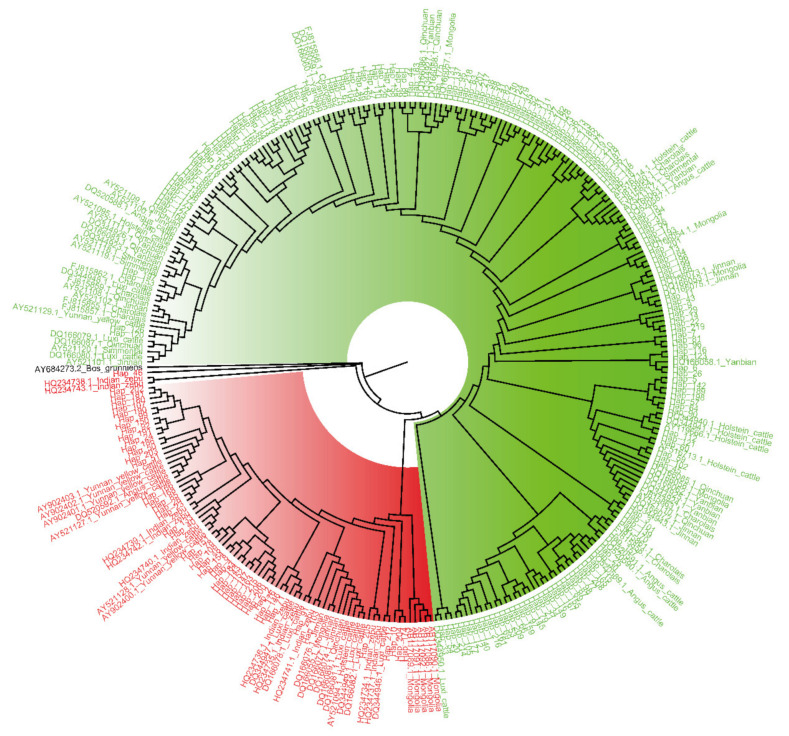
Neighbor-joining (NJ) consensus phylogenetic tree of the 219 haplotypes (red, HG1; green, HG2; black, *Bos grunniens*) and homologous sequences of other breeds from NCBI database.

**Figure 4 animals-12-02350-f004:**
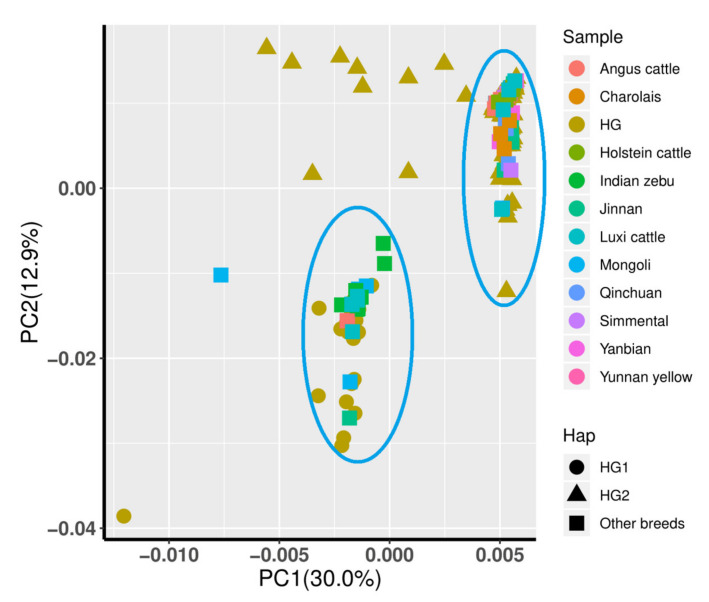
Principle components analysis (the top two components were shown) of two haplotypes and homologous sequences of other breeds from the NCBI database. Haps mean different haplogroups of Chinese Holstein cows or other breeds and are shown in different shapes. Sample representing different breeds or haplogroups are shown in different colors.

**Table 1 animals-12-02350-t001:** The mtDNA D-loop sequences downloaded from the NCBI database were used for multi-sequence alignment and NJ tree construction.

Breeds	Location	Number
Angus cattle	Western countries	6
Charolais	Western countries	10
Chinese Holstein cows	Everywhere in China	7
Indian zebu	India	10
Jinnan cattle	Middle area in China	10
Luxi cattle	Northern China	10
Mongolian cattle	Northern China	10
Qinchuan cattle	Middle area in China	10
Simmental	Western countries	4
Yanbian	Northern China	10
Yunnan yellow cattle	Southern China	7
Total	-	94

**Table 2 animals-12-02350-t002:** Association analysis of antral follicle count with different haplogroups.

	N	HG1	HG2	*p*-Value
Antral follicle count	115	1.55 ^a^ ± 0.15 (*n* = 29)	1.16 ^b^ ± 0.04 (*n* = 86)	0.014

**Note:** Antral follicles (diameter ≥ 8 mm). N/n, the number of individuals. Values with different symbols (^a^, ^b^) differ significantly at *p* < 0.05.

## Data Availability

The data presented in this study are openly available in FigShare at https://doi.org/10.6084/m9.figshare.19624548.v1.

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
