# Peer review of "Diversity of Mitochondrial DNA Haplogroups and Their Association with Bovine Antral Follicle Count"

_animals, 2022, doi:10.3390/ani12182350_

Round 1
Reviewer 1 Report
The work by Liu et al is a pleasure to read. By mtDNA sequencing, authors reveal the genetic diversity in the Chinese Holstein population, linking them to the Bos Taurus (T haplogroup) and Bos Indicus (I haplogroup). Furthermore, their suggest an association between the haplogroup type with the reproductive feature (i.e., antral follicle count). The work provides data to answer an intriguing question with regard to the genetic diversity of the Chinese Holstein, which are a result of breeding of imported Holstein cattle/semen with Chinese domestic cattle.
I only have some minor comments.
(1) Figure 2 and 3, can authors improve the “readability” of these two figures?
(2) Table 1, title line, please indicate that these sequences were downloaded and used for multi-sequence alignment and construction of NJ tree. Also, remove “statistics”.
(3) Table 2, n=29 for HG1 & n=86 for HG2. What about the other ones (the total n is 501)? If these are the number of animals that have antral follicle, 86/374=23% for HG2, vs 29/127=23% for HG1, these two have identical percentage. I think this is worth a discussion. Another question related to this is that, how many antral follicles are expected per cycle for a cow? How to explain the difference between the HG types? Does 1.55 suggest a higher possibility of twinning?
(4) Line 25, change to “cows”.
(5) Line 53, 16,338 bp. Shouldn’t this be a range? Authors found there are indel mutations, which will cause bp number change.
Author Response
[Comment 1] Figure 2 and 3, can authors improve the “readability” of these two figures?
[Response 1] For Figure 2 and 3, we enlarged the font of node labels and adjusted the highlight color, which make the figures more readability. Of which, the labels at the branches was removed to be clearer for figure 3.
[Comment 2] Table 1, title line, please indicate that these sequences were downloaded and used for multi-sequence alignment and construction of NJ tree. Also, remove “statistics”.
[Response 2] Thanks for your suggestion. The title of Table 1 was changed into “Table 1. The mtDNA D-loop sequences downloaded from the NCBI database was used for multi-sequence alignment and NJ tree construction.” In line 129-130.
[Comment 3] Table 2, n=29 for HG1 & n=86 for HG2. What about the other ones (the total n is 501)? If these are the number of animals that have antral follicle, 86/374=23% for HG2, vs 29/127=23% for HG1, these two have identical percentage. I think this is worth a discussion. Another question related to this is that, how many antral follicles are expected per cycle for a cow? How to explain the difference between the HG types? Does 1.55 suggest a higher possibility of twinning?
[Response 3] The number of animals that have antral follicle is larger than 115, but we use the interaction of valid records for all reproductive traits due to the covariate corrected model used (line 161-162). Although only one antral follicle can release to the oviduct, there are generally more than one AF in ovary. In Discussion, we discuss the potential molecular mechanism behind the difference between haplogroups and of course (last two paragraphs in discussion), but it still needs be further explored. The individuals of HG1 with the AFC of 1.55 only implies that there is higher possibility of twinning, but the application significance on IVP may be more feasible for this potential reproductive performance of HG1 due to the cattle as monotocous animal. (line 266-269)
[Comment 4] Line 25, change to “cows”.
[Response 4] We have edited it into “cows” in line 25.
[Comment 5] Line 53, 16,338 bp. Shouldn’t this be a range? Authors found there are indel mutations, which will cause bp number change.
[Response 5] This length was referenced to the length of complete mitochondrial genome (NC_006853.1) from the NCBI database in line 53.
Reviewer 2 Report
1. The chinese holstein breed is a crossbred animal. Is there a breed standard for the Chinese Holstein? Is crossing with local breeds still a regular occurrence?
2.There is mention of antral follicle count being correlated with reproductive performance? Reproductive performance is a broad term, what traits specifically is it correlated with?
3.Why are two haplotype mutations considered important in selection of a highly environmentally influenced and lowly heritable group of traits? How much variation for many reproductive traits do these two mutations account for?
4. There are 501 animals reported in this study, why were only 115 reportedly tested for HG1 and HG2?
5. The authors make reference to increased antral follicle count being greater in indicus populations and thus potential for increased reproductive performance. Could this be due to indicus animals sexually maturing later than bos taurus breeds and not pulling from their follicular pull until later in life? Furthermore, a 5-6 year old indicus animal who has likely been culled due to reproductive decline has less time being reproductivly competent than a taurus animal.
6. Is a group of culled 5-6 year old dairy cows who have been culled due to reproductive failure the best model to determine reproductive competency?
Author Response
[Comment 1] The chinese holstein breed is a crossbred animal. Is there a breed standard for the Chinese Holstein? Is crossing with local breeds still a regular occurrence?
[Response 1] Yes, the updated national breed standard of Chinese Holstein cow has been released since 2008, which stipulates the standard at breed characteristics, milk performance, reproduction performance, and adaption to local environment etc. Nowadays, the breeding targets of Chinese Holstein cows are further improvement of production performance and adaptation through crossbred with foreign or domestic breeding oxen, but improving the production performance still dominates breeding industry of Chinese Holstein cows through introducing semen of foreign breeding oxen.
[Comment 2] There is mention of antral follicle count being correlated with reproductive performance? Reproductive performance is a broad term, what traits specifically is it correlated with?
[Response 2] Ovarian function can be assessed by the AFC, and for example, the oocyte quality is associated with the AFC in cattle. Besides, AFC is also one of important indicators of ovarian reserve, which is helpful to application of assisted reproduction techniques. In terms of reproductive performance, many traits are involved in this term and AFC is only one of potential markers. Meanwhile, we added some related references (as follows) to support that in line 97.
References
- Martinez, M.F.; Sanderson, N.; Quirke, L.D.; Lawrence, S.B.; Juengel, J.L. Association between antral follicle count and reproductive measures in New Zealand lactating dairy cows maintained in a pasture-based production system. Theriogenology 2016, 85, 466-475.
- Mossa, F.; Walsh, S.W.; Butler, S.T.; Berry, D.P.; Carter, F.; Lonergan, P.; Smith, G.W.; Ireland, J.J.; Evans, A.C. Low numbers of ovarian follicles >/=3 mm in diameter are associated with low fertility in dairy cows. J. Dairy Sci. 2012, 95, 2355-2361.
- Koyama, K.; Koyama, T.; Sugimoto, M. Repeatability of antral follicle count according parity in dairy cows. J Reprod Dev 2018, 64, 535-539.
[Comment 3] Why are two haplotype mutations considered important in selection of a highly environmentally influenced and lowly heritable group of traits? How much variation for many reproductive traits do these two mutations account for?
[Response 3] The mutations mean variants distinguished between two haplogroups rather than just two haplogroups. It’s still uncertain that the effect of these mutations on AFC, but we discuss their potential molecular mechanism that these mutations may regulate the AFC through affecting mtDNA copy number. Of course, in this study, we only found this association of variants with AFC, while could not exclude the influence of other variants at mitogenome or nuclear genes. Therefore, the actual effect of variants may be relatively weaker (line 304-306).
[Comment 4] There are 501 animals reported in this study, why were only 115 reportedly tested for HG1 and HG2?
[Response 4] The number of animals that have antral follicle is larger than 115, but we use the interaction of valid records for all reproductive traits due to the covariate corrected model used, leading to the less samples with valid AFC record in statistical analysis. Relevant description of method (as follows) is in line 161-162.
Meanwhile, the research subject was changed to specific individuals with all valid records (n = 115) for achieving high confidence results.
[Comment 5] The authors make reference to increased antral follicle count being greater in indicus populations and thus potential for increased reproductive performance. Could this be due to indicus animals sexually maturing later than bos taurus breeds and not pulling from their follicular pull until later in life? Furthermore, a 5-6 year old indicus animal who has likely been culled due to reproductive decline has less time being reproductively competent than a taurus animal.
[Response 5] Considering the samples were all 5-6 years old, delay of puberty (8-14 months) should not affect the reproductive performance for Bos indicus because of passing the puberty. For second comment, Chinese Holstein cows strictly belong to Bos tarurs, but its actual genetic background tend to be complex and abundant due to the various local breeds crossbred with foreign Holstein cows. Therefore, we carry out the comparison of subpopulations with different maternal ancestries (Bos tarurs and Bos indicus) in study population (Chinese Holstein cows) rather than to analyze the divergence between different breeds (e.g. Angus and Zebu cattle) (line 106).
[Comment 6] Is a group of culled 5-6 year old dairy cows who have been culled due to reproductive failure the best model to determine reproductive competency?
[Response 6] First, the low fertility or reproduction failure is not the major reason for elimination in study population. Besides, there are various possibilities to cause the reproduction failure, including nutrition issues, diseases (not related to ovaries) etc. Meanwhile, the development of most ovaries is normal during the collection and measurement process. Therefore, it’s appropriate to determine reproductive traits using these samples (line 107).
Reviewer 3 Report
This study might give an important insight. However, I have a major concern about the materials and method.
1) Please clalify the pedigree or family structure and explain the validity of how to collect samples used.
2) Please explain the validity of the statistical model, expecially in terms of controlling the effect of genes on nuclear genome.
Author Response
[Comment 1] Please clarify the pedigree or family structure and explain the validity of how to collect samples used.
[Response 1] We are sorry to cannot provide the corresponding family structure and it is also not involved in this statistical model. However, this study itself aims to find the subpopulation structure (haplogroup) and evaluate its association with reproductive traits in all individuals, which may account for the potential effect of family structure on reproductive traits. (line 269-271)
[Comment 2] Please explain the validity of the statistical model, especially in terms of controlling the effect of genes on nuclear genome.
[Response 2] Honestly, the effect of genes on nuclear genome was not involved in statistical model. However, due to the random sampling process, the genetic background on nuclear genome of study individuals may be similar or its effect is weak between two haplogroups.
Round 2
Reviewer 3 Report
I strongly recommend to add some sentences to declare that this study assumed that the sampling was at random about pedigree/family relatedness and there was no bias about the frequency of other genes.
Author Response
[Comment 1] I strongly recommend to add some sentences to declare that this study assumed that the sampling was at random about pedigree/family relatedness and there was no bias about the frequency of other genes.
[Response 1] Thanks for your suggestion. We emphasized the sampling process was random so that the potential effect of family structure and other genes were ignored in this study (line 106-109).